# Application of Hyperspectral Remote Sensing in the Longwave Infrared Region to Assess the Influence of Dust from the Desert on Soil Surface Mineralogy

**Gila Notesco \*, Shahar Weksler and Eyal Ben-Dor** 

Porter School of the Environment and Earth Sciences, Faculty of Exact Sciences, Tel Aviv University, Tel Aviv 6997801, Israel; weksler@mail.tau.ac.il (S.W.); bendor@tauex.tau.ac.il (E.B.-D.)
\* Correspondence: gilano@tauex.tau.ac.il

**Abstract:** Soil mineralogy can be used to study changes in the environment affecting the soil surface, such as dust from the desert through Aeolian processes, which is one of the sources that determine the mineral nature of the soil. Ground- and field-based hyperspectral longwave infrared images, acquired before and after dust dispersion on the soil surface, were processed and analyzed by applying a procedure for determining soil surface mineralogy from the emissivity spectrum, using two indices—SQCMI (the Soil Quartz Clay Mineral Index) and SCI (the Soil Carbonate Index)—to identify changes in the abundance of quartz, clay minerals and carbonates on the surface, caused by the settling dust particles. Mineralogical changes were identified, depending on the mineral composition of the dust compared to the soil surface mineralogy.

**Keywords:** hyperspectral remote sensing; longwave infrared image; emissivity spectrum; soil mineralogy; desert dust

## 1. Introduction

Soil mineralogy, which holds important information on soil origin and development, can be used to study changes in the environment affecting the chemo-physical properties of the surface. Dust from the desert, through Aeolian processes, is one of the sources that determine the nature of the soil; the settling dust particles are one of the major sources of soil mineral nutrients, especially in arid and semiarid regions [1,2]. The most common minerals in soils, as well as in desert dust particles, quartz, clay minerals, and carbonates, present with fundamental spectral features in the thermal infrared, mainly in the longwave infrared (LWIR, 8–12 μm) region, due to the fundamental vibration modes of the silicon–oxygen bond (Si–O) in quartz and clay minerals, and the carbon–oxygen bond (C–O) in carbonates. Hyperspectral remote sensing in the LWIR region, a useful tool for mineral mapping [3–8], can be used to determine the content of these minerals in the soil surface and identify changes in their abundance. In a previous study [9], we developed a procedure for determining soil mineralogy using LWIR images. The emissivity spectra of tens of soil samples were calculated and analyzed to identify mineral-related emissivity features and their relative intensities. Two created indices—SQCMI (Soil Quartz Clay Mineral Index), indicating the amount of quartz relative to clay minerals, and SCI (Soil Carbonate Index), indicating the concentration of carbonates in the soil—enabled us to determine the mineralogy, from more to less abundant, in each soil sample. Pursuant to those results, we applied the procedure to ground- and field-based hyperspectral LWIR images to study the ability to identify changes in the abundance of quartz, clay minerals and carbonates in the soil resulting from the dispersion of desert dust-like materials on the surface, and to assess the interaction of the soil surface with desert dust particles.

## 2. Materials and Methods

Ground-based images of soil samples, collected from the surface (0–5 cm depth) at different sites in Israel, were acquired with the Telops Hyper-Cam sensor [10], located at a distance of about 2 m from the samples, covering the LWIR spectral region (8.0–11.7 μm) with 122 bands and a spectral resolution of 4 cm$^{-1}$. After a first LWIR image was acquired (Figure 1), desert loess soil, a sediment formed by the accumulation of Aeolian dust particles (e.g., [11]), was dispersed on the surface of each soil sample and a second LWIR image was acquired. The emissivity spectrum of each soil sample was calculated as described in [9], and then the SQCMI and SCI values of each sample, without and covered with the desert loess soil (henceforth dust_1), were calculated according to Equations (1) and (2). These values were used to study whether and how the settled particles affect the mineral-related spectral features and, therefore, the determination of quartz, clay minerals, and carbonates contents in each soil sample.

$$SQCMI = N\varepsilon_{\lambda = 9.56\ \mu m}/(N\varepsilon_{\lambda = 8.21\ \mu m} \times N\varepsilon_{\lambda = 8.85\ \mu m}) \tag{1}$$

and,

$$SCI = N\varepsilon_{\lambda = 11.24\ \mu m} \times N\varepsilon_{\lambda = 10.51\ \mu m}/N\varepsilon_{\lambda = 8.85\ \mu m} \tag{2}$$

where $N\varepsilon$ is the normalized emissivity value at the indicated wavelength ($\lambda$). In general, a larger SQCMI value indicates a higher amount of quartz relative to clay minerals; a smaller SCI value indicates a higher concentration of carbonates in the soil sample [9]. In addition, two other desert loess soils (henceforth dust_2 and dust_3) and dune sand, with different mineral contents, were dispersed, separately, on the surface of a loess soil sample.

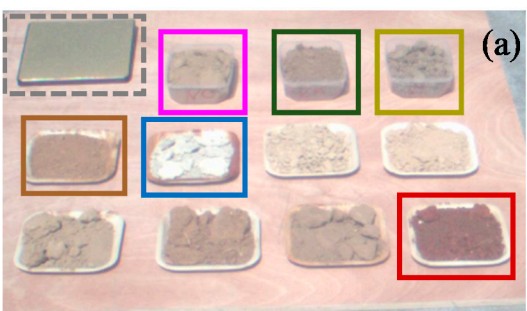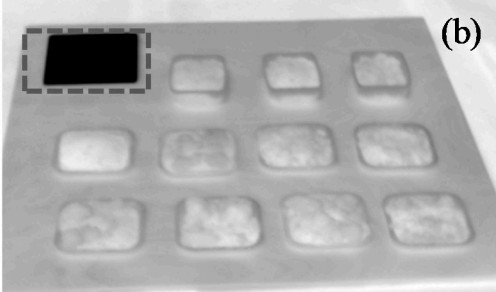

**Figure 1.** Red-green-blue (RGB) image (**a**) and longwave infrared (LWIR) image (band 10.62 μm) (**b**) of soil samples (each placed on a 0.04 m$^2$ plate) and a gold plate (gray dashed square), measuring the downwelling radiance (used to calculate the emissivity spectrum of each soil sample, as described in [9]); emissivity spectra of selected soil samples (colored squares) are shown in the next section (with respective colors).

To identify regional-scale mineralogical changes on the soil surface, field campaigns were conducted in two study areas, consisting of agricultural fields with different soil types (Figure 2). In each area, after a first LWIR image was acquired, desert dust-like materials (henceforth dust) were dispersed on the surface, and then a second LWIR image was acquired and the emissivity spectrum of each pixel was calculated, resulting in before- and after-dust-dispersion emissivity images. Then, the SQCMI and SCI values of each pixel in each image were calculated as described above, resulting in before- and after-dust-dispersion SQCMI and SCI images.

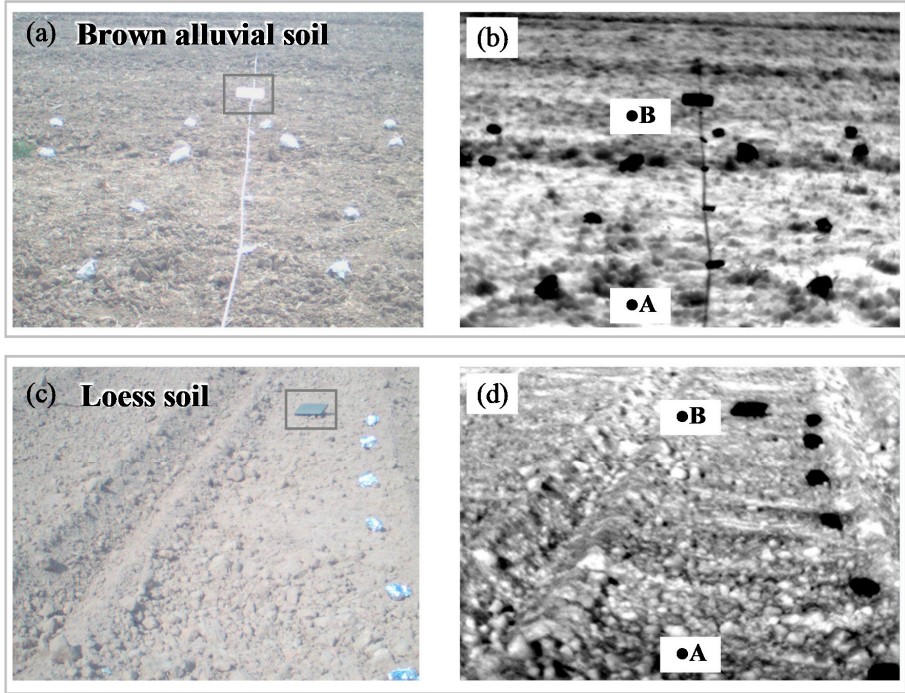

**Figure 2.** (**a**) and (**c**) RGB images of the two study areas; gold plate (gray square) and aluminum foil signs (delimiting 1-m$^2$ regions in which dust-like materials were dispersed) were placed in each area. (**b**) and (**d**) Respective LWIR images (band 10.62 µm); the sensor was located at a distance of 3 and 8 m from points A and B, respectively, resulting in a spatial resolution of 0.5 and 1 cm, respectively.

## 3. Results and Discussion

### 3.1. Ground-Based Data

Calculated emissivity spectra of the soil samples are shown in Figure 3; the resulting index values and mineralogy of each soil sample and dispersed dust are given in Table 1.

**Table 1.** Spectral indices and mineralogy of soil samples and dispersed dust.

| Soil Sample/Type | SQCMI | SCI | Mineralogy (More to Less Abundant) [1] | |
|---|---|---|---|---|
| 1/Loess | 1.032 | 1.009 | Q C CM | |
| 2/Brown alluvial_1 | 1.007 | 1.011 | CM Q C | |
| 3/Brown alluvial_2 | 1.010 | 1.006 | Q C CM | |
| 4/Brown steppe | 1.003 | 1.005 | CM C Q | |
| 5/Brown desert skeletal | 0.997 | 1.000 | C CM Q | |
| 6/Terra rossa | 0.956 | 1.018 | CM Q C | |
| **Dispersed dust** | | | | |
| Dust_1 | 1.015 | 0.998 | C Q CM | |
| Dust_2 | 1.026 | 1.005 | Q C CM | Increasing Q, |
| Dust_3 | 1.125 | 1.072 | Q CM | decreasing C |
| Dune sand | 1.453 | 1.278 | Q | ↓ |

[1] Determined according to [9]; Q, quartz; CM, clay minerals; C, carbonates. SQCMI (Soil Quartz Clay Mineral Index). SCI (Soil Carbonate Index).

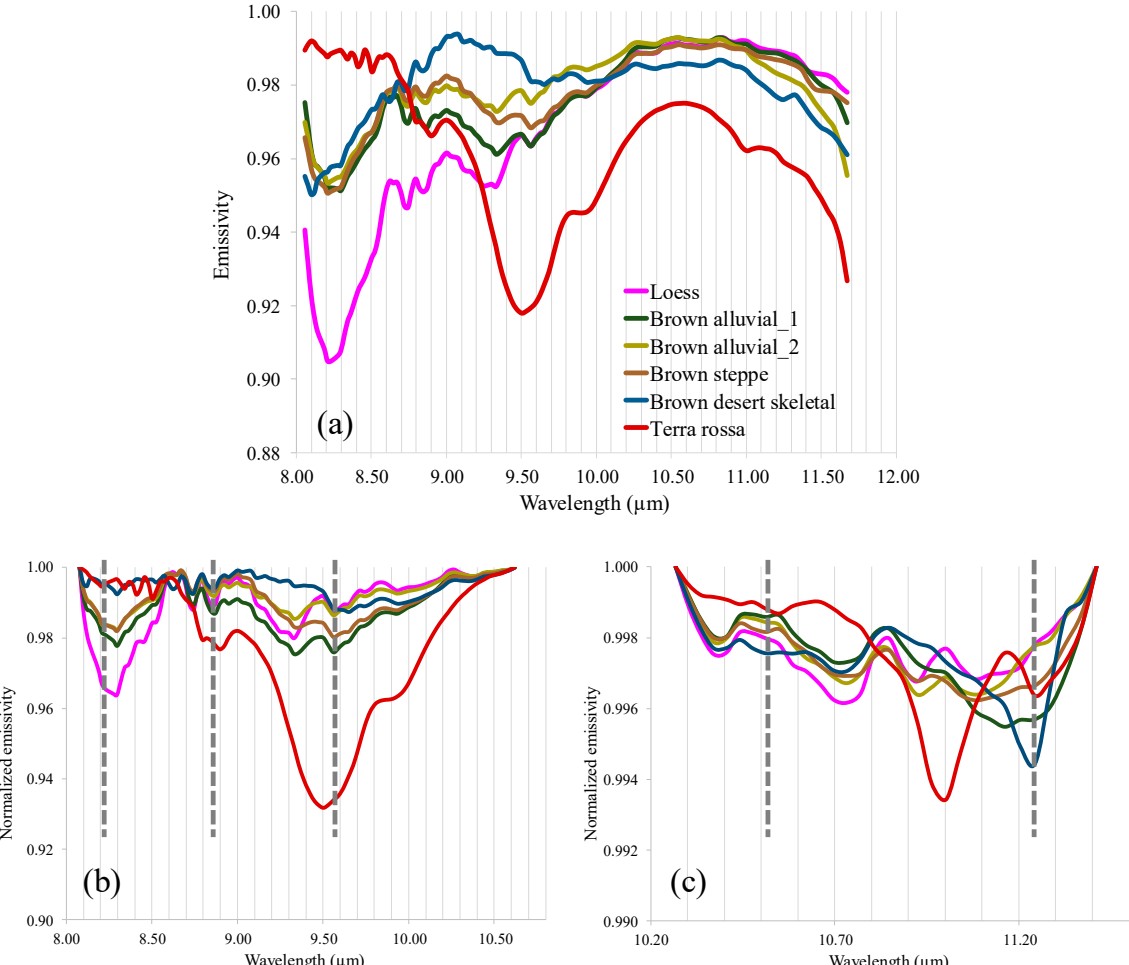

**Figure 3.** (**a**) Emissivity spectra of soil samples; each spectrum is the average of tens to hundreds of pixels in the image. (**b**) and (**c**) Normalized emissivity spectra (continuum removal normalization in specific spectral ranges [12]), of the soil samples emphasizing mineral-related absorption features of quartz and clay minerals (**b**) and carbonates (**c**), as described in [9]; gray dashed lines indicate the wavelengths mentioned in Equations (1) and (2).

The effect of settling dust_1 on the soil surface mineralogy is shown in Figure 4. The addition of carbonates-rich dust_1 (lowest SCI value, Table 1) resulted in decreased SCI values, expressing the increase in the amount of carbonates, in all soil samples except for the carbonate-rich 5/brown desert skeletal soil (Figure 4b). Changes in the amount of quartz relative to clay minerals were noticed mainly in the clay minerals-rich 6/terra rossa soil (increased SQCMI value, indicating an increase in the relative amount of quartz), and in the quartz-rich 1/loess soil (decreased SQCMI value, indicating a decrease in the relative amount of quartz). The addition of quartz-rich dust_2 to the surface of 1/loess soil resulted in similar, but smaller, mineralogical changes compared to the addition of dust_1, expressed by negative ΔSQCMI and ΔSCI values (Figure 5), where Δ is the difference between after and before the dust dispersion. On the other hand, addition of dust_3 or dune sand, which are very rich in quartz (Table 1), to the surface of 1/loess soil resulted in an increase in the relative amount of quartz and a decrease in the amount of carbonates, expressed by positive ΔSQCMI and ΔSCI values (Figure 5). All in all, the identification of mineralogical changes depended on the mineral composition of the dust compared to the soil surface mineralogy.

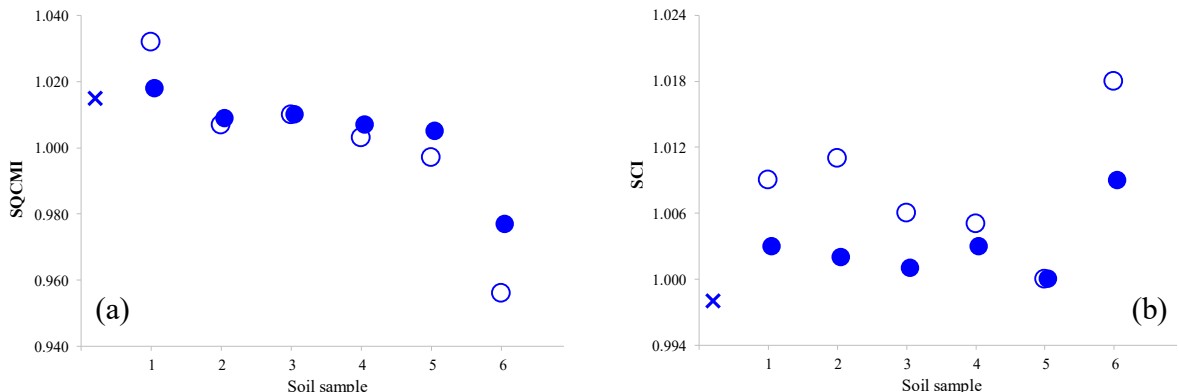

**Figure 4.** Soil Quartz Clay Mineral Index (SQCMI) (**a**) and Soil Carbonate Index (SCI) (**b**) values of soil samples: without dust_1 (empty circles), and covered with 50 g m$^{-2}$ dust_1 (filled circles); x represents the index values of dust_1.

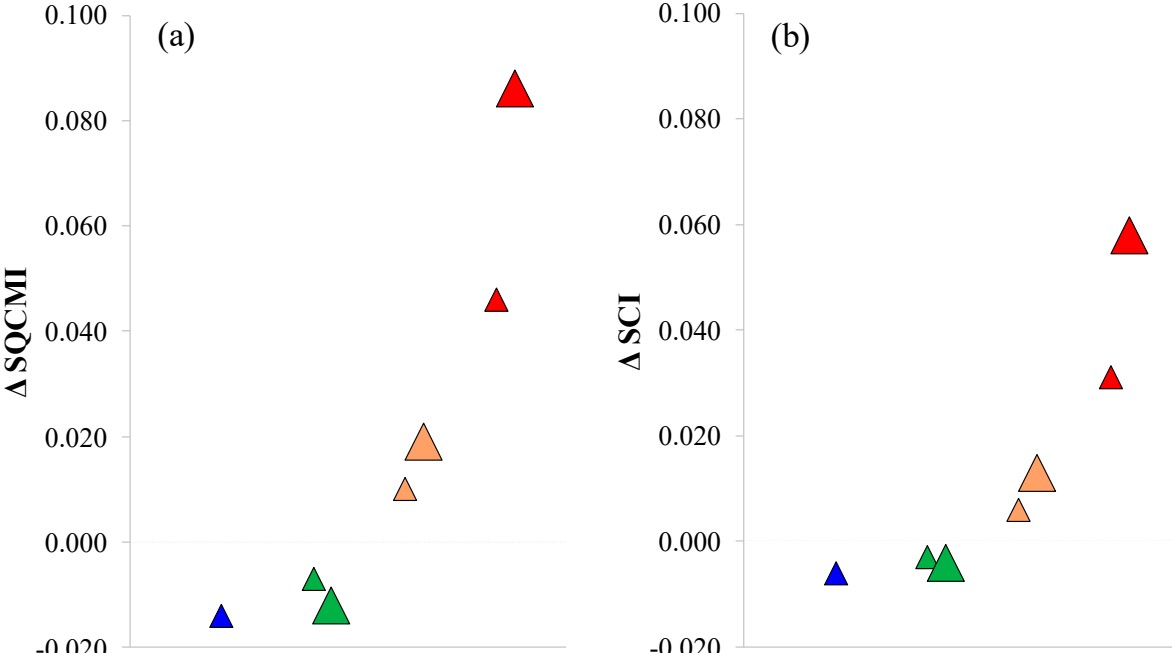

**Figure 5.** ΔSQCMI (**a**) and ΔSCI (**b**) values resulting from the dispersion of 50 g m$^{-2}$ (small triangles) and 100 g m$^{-2}$ (big triangles) of dust_1 (blue), dust_2 (green), dust_3 (orange) and dune sand (red) on the surface of 1/loess soil; *x*-axis offset for clarity.

### 3.2. Field-Based Data

The mineral composition of the soil surface in each study area, as determined from the emissivity image before dust dispersion, is given in Table 2. Dividing the after-dust-dispersion emissivity image by the before-dust-dispersion image and applying the spectral angle mapper (SAM) algorithm [13,14] to the resultant divided image, with a unity spectrum as the endmember spectrum (representing "no change" between the two images), enabled detecting changes on the surface (Figures 6a and 7a,b) resulting from settling of the dust particles. Note that the addition of dust_1, with different mineralogy than the brown alluvial soil, is more noticeable than the addition of dust_2, despite the former's smaller amount (50 g m$^{-2}$ and 250 g m$^{-2}$, respectively, Figure 6a). On the other hand, the addition of dust_2 to the surface of the loess soil, with different mineralogy, is noticeable (Figure 7a, b). Once a change was detected, its mineral nature was determined by comparing the index values after and before dust dispersion, giving ΔSQCMI and ΔSCI images (Figure 6b,c and Figure 7c–f). The addition of dust_1

to the surface of the brown alluvial soil resulted in negative ΔSCI values, indicating an increase in the amount of carbonates (Figure 6c). The addition of dust_2 to the surface of the loess soil resulted in negative ΔSQCMI and ΔSCI values, indicating a decrease in the relative amount of quartz and an increase in the amount of carbonates, respectively, (Figure 7c–f), whereas the addition of dust_3 or dune sand resulted, in both study areas, in positive ΔSQCMI and ΔSCI values, indicating an increase in the relative amount of quartz and a decrease in the amount of carbonates, respectively.

**Table 2.** Spectral indices and mineralogy of soil surface.

| Study Area Soil Type | SQCMI [1] | SCI [1] | Mineralogy (More to Less Abundant) |
|---|---|---|---|
| Brown alluvial | 1.019 ± 0.007 | 1.003 ± 0.008 | Q C CM |
| Loess | 1.038 ± 0.010 | 1.026 ± 0.011 | Q CM C |

[1] Each value is the average ± standard deviation of all soil pixels in the image.

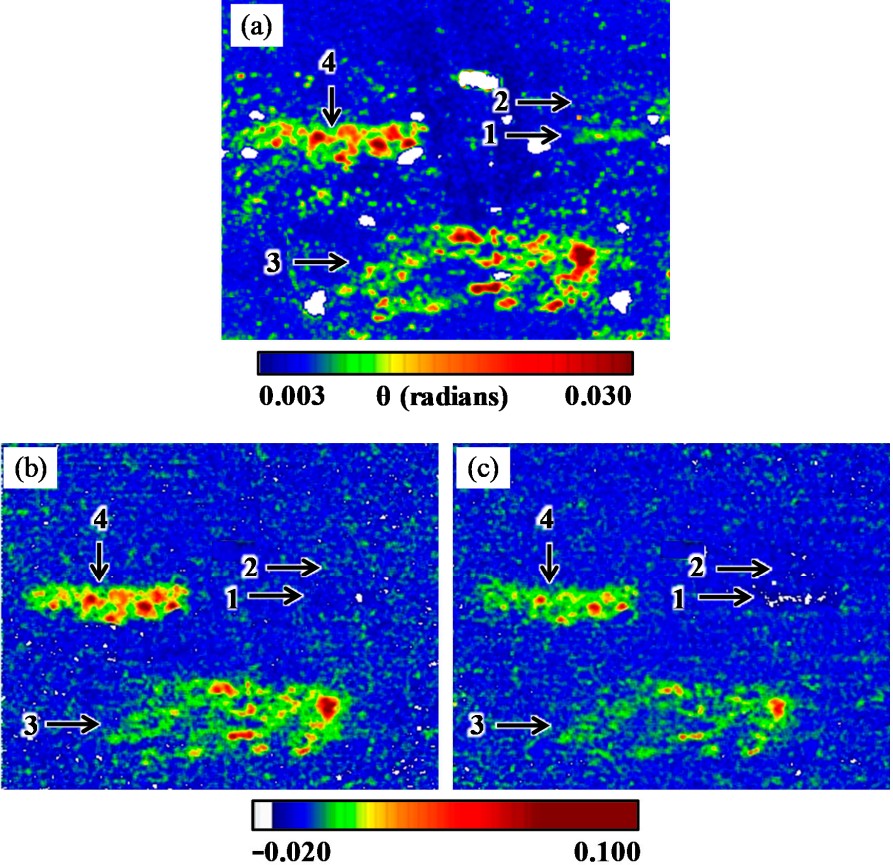

**Figure 6.** (**a**) Spectral angle mapper (SAM) algorithm rule map of the brown alluvial soil study area emphasizing changes on the surface resulting from the dispersion of 50 g m$^{-2}$ dust_1 (region 1), 250 g m$^{-2}$ dust_2 (region 2), 250 g m$^{-2}$ dust_3 (region 3) and 250 g m$^{-2}$ dune sand (region 4); a larger angle (θ) indicates bigger changes on the surface. ΔSQCMI (**b**) and ΔSCI (**c**) images; blue color in (**b**) and (**c**) represents ΔSQCMI and ΔSCI ≈ 0, i.e., no mineralogical changes. The non-homogeneity of θ, ΔSQCMI and ΔSCI values within a region, resulting from the dispersion procedure, emphasizes the sensitivity for detecting local mineralogical changes in the soil surface.

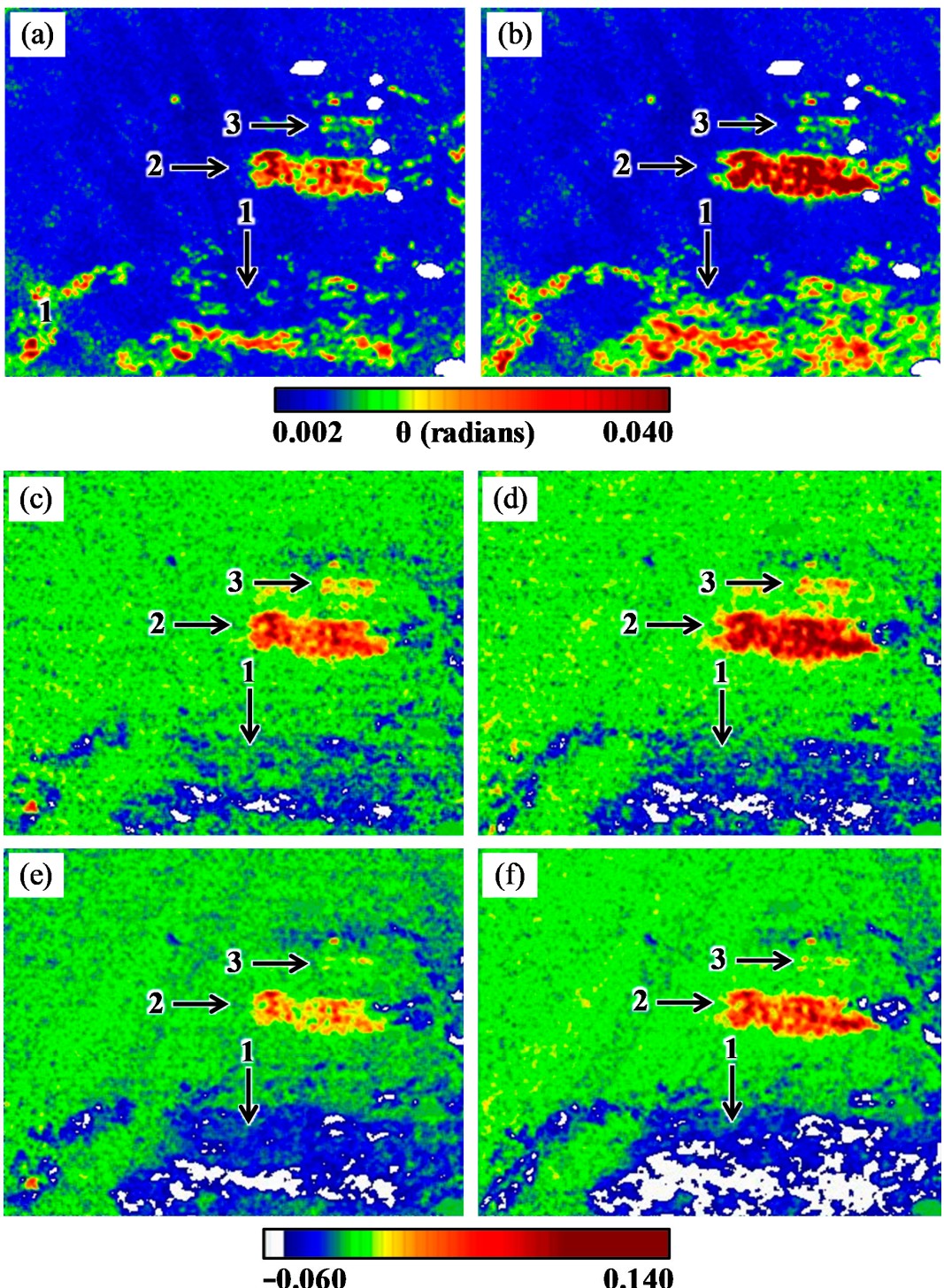

**Figure 7.** SAM algorithm rule map (**a**), ΔSQCMI image (**c**) and ΔSCI image (**e**) of the loess soil study area emphasizing changes on the surface resulting from the dispersion of 100 g m$^{-2}$ dust_2 (region 1), 100 g m$^{-2}$ dune sand (region 2) and 50 g m$^{-2}$ dune sand (region 3). SAM algorithm rule map (**b**), ΔSQCMI image (**d**) and ΔSCI image (**f**) resulting from the dispersion of 250 g m$^{-2}$ dust_2 (region 1), 250 g m$^{-2}$ dune sand (region 2) and 50 g m$^{-2}$ dune sand (region 3); green color in (**c**)–(**f**) represents ΔSQCMI and ΔSCI≈0, i.e., no mineralogical changes.

## 4. Conclusions

We demonstrate the ability to identify mineralogical changes in the soil resulting from dispersion of desert dust-like materials on the surface, applying a procedure that we developed previously [9]. Ground- and field-based hyperspectral LWIR images, acquired before and after dust dispersion on the soil surface, were used to calculate the SQCMI and SCI values and identify mineralogical changes in the amount of quartz relative to clay minerals, and the amount of carbonates in the soil surface. The identification of mineralogical changes depended on the mineral composition of the dust compared to the soil surface mineralogy, and the amount of settling dust.

The mapping of soil mineral abundance and monitoring of mineralogical changes resulting from settled desert dust aerosols are of interest on a regional scale. Acquiring field and/or airborne LWIR images, before and after a significant dust event, and applying the developed procedure, will yield the best assessment of the interaction of the soil surface with desert dust aerosols. Moreover, the procedure can be used to study the mineralogical effects of other processes, natural and unnatural, affecting the soil surface.

**Author Contributions:** Conceptualization, G.N. and E.B.-D.; Methodology, formal analysis, validation, writing—original draft preparation and editing, G.N.; resources, writing—review, supervision, E.B.-D.; Sensor operation and data acquisition, S.W. All authors have read and agreed to the published version of the manuscript.

**Funding:** This research was funded by the Israel Ministry of Science, Technology and Space, grant number 68740.

**Conflicts of Interest:** The authors declare no conflict of interest.

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
