# Peer review of "Application of Hyperspectral Remote Sensing in the Longwave Infrared Region to Assess the Influence of Dust from the Desert on Soil Surface Mineralogy"

_remotesensing, doi:10.3390/rs12091388_

Round 1

Reviewer 1 Report

This technical note proposes a method to identify changes in the abundance of quartz, clay minerals and carbonates in soil surface from the emissivity spectrum by using SQCMI and SCI indexes.

In my opinion the method is interesting, and the results section is convincing. Nonetheless the manuscript presents one weakness: Introduction section is too brief and should include more information on state-of-the-art, even for a technical note. They still have one page to use.

Hence: I recommend to improve the Introduction; a proper one should introduce the context in which the problem is relevant, cite the proper sources, report the history and the main steps in the related research by highlighting the limits and by explaining how the proposed methods can face them.

I suggest firstly discuss about the hyperspectral imaging applied to different but similar problem like terrain classification from aerial view. Here some papers to consider:

  1. Devaram, R. R., Allegra, D., Gallo, G., Stanco, F.: Hyperspectral Image Classification via Convolutional Neural Network Based on Dilation Layers. International Conference on Image Analysis and Processing, in Lecture Notes in Computer Science, volume 11751 (2019).
  2. Gao, F., Wang, Q., Dong, J., Xu, Q.: Spectral and spatial classification of hyperspectral images based on random multi-graphs. Remote Sensing 10, 1271 (2018).
  3. Li, Y., Zhang, H., Shen, Q.: Spectral-spatial classification of hyperspectral imagery with 3D convolutional neural network. Remote Sensing 9(1) (2017).

Then the authors must focus on works which address the same tasks faced in their technical note, in order to highlight the advantages of the proposed study. Here some examples:

  1. Keller, S., Riese, F. M., Stötzer, J., Maier, P. M., and Hinz, S.: Developing a Machine Learning Framework for Estimating Soil Moisture with Vnir Hyperspectral Data, ISPRS Ann. Photogramm. Remote Sens. Spatial Inf. Sci., IV-1, 101–108 (2018)
  2. Omran, E.S.E. Rapid prediction of soil mineralogy using imaging spectroscopy. Eurasian Soil Science 50, 597–612 (2017).
  3. S. Galvão, A. R. Formaggio, E. D. Couto, and D. A. Roberts, “Relationships between the mineralogical and chemical composition of tropical soils and topography from hyperspectral remote sensing data,” ISPRS J. Photogramm. Remote Sens. 63, 259–271 (2008).

Overall, the manuscript is average and could be accepted after a proper major revision which addresses the aforementioned comments.

Author Response

Our reply is given in red color below.

I recommend to improve the Introduction; a proper one should introduce the context in which the problem is relevant, cite the proper sources, report the history and the main steps in the related research by highlighting the limits and by explaining how the proposed methods can face them.

A few sentences were added to the introduction, including some appropriate references, relevant to hyperspectral remote sensing in the LWIR spectral region.

Reviewer 2 Report

Dear authors, I sincerely believe that the article needs a thorough review of each and every one of its sections. What is to be explained is not well understood, the procedures used are not explained, nor are the figures shown. It is very confusing, badly structured and therefore the substance of what is wanted to be explained by the forms used is lost.

The introduction is practically inexistent, I think it should be completed and extended as well as the only 6 bibliographical references used in the text

Figure 1 needs further explanation of what each sample is shown for. Also, the paragraph above talks about a capture before and after dispersing the dust, but there is no before and after image. And later it talks about dispersing more types of dust. I think it would be interesting to show the images used as a whole, before and after the dispersion of the different types of dust used. And to indicate how the different types of dust are dispersed and how they are removed in order to disperse others, if that is the procedure.

Similarly in figure 2 an example of two study areas used is shown, but only before the dispersion of the powder, I think that images should be shown before and after the dispersion of the powder, or at least of the different areas through which the dispersion of the different types of powder is to be carried out, as seems to be deduced from figures 6 and 7. And to clarify the relationship between figure 2 and figure 1, as it is not clear whether the samples in figure 1 are those obtained in the study areas of figure 2 or not, or which.

On the other hand, the role of the gold and aluminium references in the different images is not explained. What exactly are they used for? Because only gold is used in figure 1 and gold and aluminum in figure 2.

In Figure 3, how is Figure 3a obtained, how is the normalization of Figures 3b and 3c carried out, why has it been separated into two subfigures "b" and "c" by wavelength, should there not be a graph of the whole wavelength range for each of the normalizations of Equations 1 and 2?

Given the importance of the SAM algorithm, I think it should be explained in the text and not just referenced.

Figures 6 and 7 are very interesting and I think they show the potential of the article, but I think an effort should be made to better explain each of the scenarios used, the dispersion zones of each type of dust, as well as the comparative results obtained.

In general, I think it is better if you show the figures before the paragraph where the results obtained from your study are explained.

The conclusions are too general, I think that more specific conclusions to the results obtained should be shown.

Author Response

Our reply is submitted as a PDF file.

Round 2

Reviewer 1 Report

The authors have properly improved the manuscript by making stronger the introduction.

In my opinion the TN is now ready for the publication.

Reviewer 2 Report

I would like to thank the authors for the changes made to the text. I had mistakenly judged it to be an article and not a technical note, and that is why I felt it was too short. In any case, most of the considerations made have been taken into account and I believe that it is now much clearer and easier for the reader to follow.